# Air Pollution Modeling for Sustainable Urban Mobility with COVID-19 Impact Analysis: Case Study of Skopje

Mladen Miletić *, Edouard Ivanjko, Tomislav Fratrović and Borna Abramović

Faculty of Transport and Traffic Sciences, University of Zagreb, Vukelićeva 4, 10000 Zagreb, Croatia
* Correspondence: mladen.miletic@fpz.unizg.hr

**Abstract:** Air pollution is one of the major problems in today's urban areas. With increasing development and the need for the transport of goods and people, it has become imperative to seek sustainable urban mobility solutions. The impact of restrictive COVID-19 pandemic measures provides a unique insight into the possible reductions in air pollution. This paper presents a case study on the city of Skopje, North Macedonia, and attempts to identify the effect of traffic emissions on air quality. Resultant correlation analysis and linear regression models show the impacts of multiple factors contributing to air pollution. Finally, a discussion on the impact of COVID-19 measures on air pollution is provided. The main findings of this research are the observed drop in air pollution levels during COVID-19 measures, the effects on air pollution models, and the identification of primary pollutants in the city of Skopje.

**Keywords:** air pollution; air quality; sustainable urban mobility; data science; COVID-19

## 1. Introduction

The mobility of goods and people is one of the major contributors to the development and prosperity of urban areas. In countries without developed sustainable public transport infrastructure, most people commute by using private vehicles, thereby causing increased levels of congestion. With increased congestion, vehicles spend more time in intersection queues with their engines turned on and emitting pollutants. With increasing pollution levels, the air quality and living conditions continuously degrade. Obvious solutions to this growing problem are investment in sustainable modes of public transport, switching to alternative fuels, and improving the existing transport infrastructure. Unfortunately, it is challenging to persuade both the public and regulatory officials to act on this matter. One obstacle is the difficulty in estimating the actual amount of air pollutants that are directly caused by traffic, as multiple other factors, such as industry and meteorological conditions, can affect the measured pollution levels. With proper assessment of traffic-induced air quality degradation, future sustainable mobility concepts can be developed. With the identification of key problem areas or problematic time intervals, possible solutions will be easier to develop.

Since the year 2019, a global pandemic of the severe acute respiratory syndrome coronavirus 2 (SARS-CoV-2) causing the coronavirus disease of 2019 (COVID-19) has severely affected all industrial and social elements of human life. Various restrictive measures have been imposed to differing degrees throughout the world to deal with the pandemic. The imposed measures range from simple recommendations to increase hygiene levels to even full national lockdowns and restrictions on movement and mobility. The full impact of both the pandemic and the imposed measures is yet to be determined, but the effect of restrictive measures on air quality in urban areas can be analyzed. This took place in the city of Skopje, North Macedonia, using existing traffic and air-quality-measurement instruments, and with highly restrictive measures on mobility. The main goal and research objective of this paper is to model air pollution with respect to traffic

flow and meteorological conditions. A research question arose on how the COVID-19-related restrictive measures impacted the air quality and what their impact was on the model's coefficients.

This paper is comprised of six sections. After the introductory section, the relevant literature is reviewed in section two. Section three explains the methodology and data used for processing, correlation, and regression analysis. In section four, the results are presented, followed by a discussion in section five. The final section concludes the paper with a commentary on possible improvements and future work.

## 2. Literature Review

In the available literature, the transport sector has long been associated with rising air pollution in urban areas [1]. The chemical properties of internal combustion engines are to blame for rising CO and $NO_x$ levels, as those gasses are byproducts of their general operation. In addition, a lot of particulate matter (PM) has also been attributed to the transport sector. Many adverse health effects are attributed to rising air pollution levels, such as respiratory problems, cardiovascular health decline, and even cancer [2]. The technological and technical development of engines eventually allowed reduced emissions, but it remains unlikely that the zero emissions goal will ever be met while combustion engines are in operation [3]. With increasing norms and regulations in developed countries, a switch toward the electrification of transport has started [4]. Unfortunately, developing countries are not only slow to introduce electric vehicles, but also, the vehicles in those countries tend to be older, generating increased emissions.

The city of Skopje, the capital of North Macedonia, which is a developing country, is known for high levels of air pollution. Multiple studies on air quality in Skopje have been conducted supporting this situation. The research in [5] identified air pollution as a major environmental health problem in Skopje. In addition to health problems, the economic and social costs of pollution were also considered to be very high. Studies of Skopje air pollution [5–7] identified two key components of pollution in Skopje. The first component is the human influence in the form of heavily congested traffic flow, residential heating systems, and industrial activity in the city. The second component is the geographical nature, since the city is located in a river-shaped valley surrounded by mountains. This has the side-effects of a low wind speed, high humidity during winter, and temperature inversions which trap air pollutants in the city.

Since the start of the COVID-19 pandemic, the lockdown's effects on air pollution have been closely monitored by researchers. The impact of COVID-19 measures in Qatar was discussed in [8], and it was observed that overall, the traffic demand was reduced by 30% while measures were active. Side-effects of this reduction the reductions in traffic violations and crashes, by 73% and 37%, respectively. In [9], the impact of COVID-19 measures on air quality in Almaty, Kazakhstan, was analyzed. The paper noted that there were substantial reductions in CO, PM$_{2.5}$, and NO$_2$ concentrations during the lockdown. However, it is noted that even in traffic-free conditions, during the lockdown, the overall air quality remained low, as there were many other sources of pollution in the city. In [10], the authors discussed a comprehensive study of private vehicle restrictions policies in 49 cities in China. A noticeable effect on air pollution was observed, and it is discussed that policies implemented for COVID-19 spread reduction could also be implemented for the purpose of air pollution improvement. The research in [11] observed reductions in congestion, mobility, and NO$_2$ during the lockdown across 22 US cities using regression models. The research was also conducted in smaller cities, as reported in [12], where the city of Maribor, Slovenia, was observed during COVID-19 lockdowns. It was reported that the reduction in NO$_2$ was smaller than the reduction in traffic volume due to a shift to more NO$_2$-dominant traffic sources, such as diesel-powered heavy-goods vehicles, which remained in operation even during the lockdown. A study in [13], which also focused on the city of Maribor, observed a similar reduction in PM particle concentration.

In the available literature, only two papers focused on the effects of COVID-19 measures on air quality in Skopje. The first paper [14] provides an early analysis of air quality state following the most restrictive part of 2020's COVID-19 measures. The results show that there was an reduction in air pollution during the lockdown period, but there is no direct comparison with traffic and meteorological data. The second paper [15] analyzed the impact of COVID-19 measures in Skopje on noise pollution levels, observing reductions. The reduced noise levels were in part caused by the reduced road traffic, but a direct correlation or further analysis was not provided.

A systematic review of COVID-19 measures on air quality presented in [16] revealed that in most study areas around the world, there was a general improvement in air quality during the imposed lockdowns. In general, it was observed that the values of $NO_2$ and $PM_{10}$ concentrations decreased significantly. The value of $O_3$ concentration usually showed a small increase. The CO levels were more diverse, depending on the study location. In [16], the authors also identified that more studies are needed, as each study location had a unique combination of policies, meteorological conditions, geography, and air pollution levels. In addition, an analysis of the imposed restrictions impact on air quality beyond the first lockdown should be conducted.

Considering the literature review above, several research gaps were found that are addressed in this paper. The first identified research gap is that most research on the impact of COVID-19 on air quality has focused on brief periods of strict lockdowns, usually at the beginning of 2020. Our study included data from two years before COVID-19 and the entirety of 2020. The second identified research gap is that there has been no analysis of how regression models, frequently used in periods before COVID-19 to model the relation of traffic and air pollution, were affected by COVID-19. In our study, a comparison is made between regression models created from data before COVID-19 measures and after. The main contribution of this paper is the analysis of the correlation between air quality and traffic flow, with a special emphasis on outlier behavior caused by COVID-19-related restrictive measures using real data collected in the city of Skopje, North Macedonia. For the analysis, the traffic, air quality, and meteorological data were collected from the beginning of 2018 until the end of 2020. While this paper deals with the impacts of COVID-19 measures on traffic and air quality, it should be noted that other effects of COVID-19 measures (political, economic, and social) are beyond the scope of this paper and will not be discussed.

## 3. Data and Methodology

In this section, the used data sets with initial observations and methodology are presented.

### 3.1. Data Sources

The data collected for the purpose of this research consisted of several datasets comprising data about traffic flow, air quality, and meteorology. In addition, characteristic dates of COVID-19-related restrictive measures in North Macedonia were also recorded.

#### 3.1.1. Traffic-Flow Dataset

The first dataset is related to the measured traffic flow through intersections in the city of Skopje obtained with the courtesy of the Centre for Traffic Management and Control—Skopje. The dataset consists of the number of vehicles at each intersection with one-hour resolution for one full week in each month of 2018, 2019, and 2020. The initial preprocessing was used to remove data on days with missing measurements. After the initial preprocessing, four intersections out of X available were selected for further analysis. The four selected intersections were chosen according to the following selection criteria: (i) measuring sensors in operation above 80% of the time; (ii) intersections on crucial city transport corridors or with significant traffic flow; (iii) the intersection's distance from the air quailty measurement station. Locations of the chosen intersections are shown in Figure 1. The measurements of intersection $I_1$ only include measurements from three

intersection inputs, as the measuring sensor for the fourth input was not operational for most of the data collection period. Summary statistics of the filtered traffic dataset are shown in Table 1. In addition to the traffic-flow dataset, the general statistics of registered road vehicles regarding their fuel type and age are shown in Table 2.

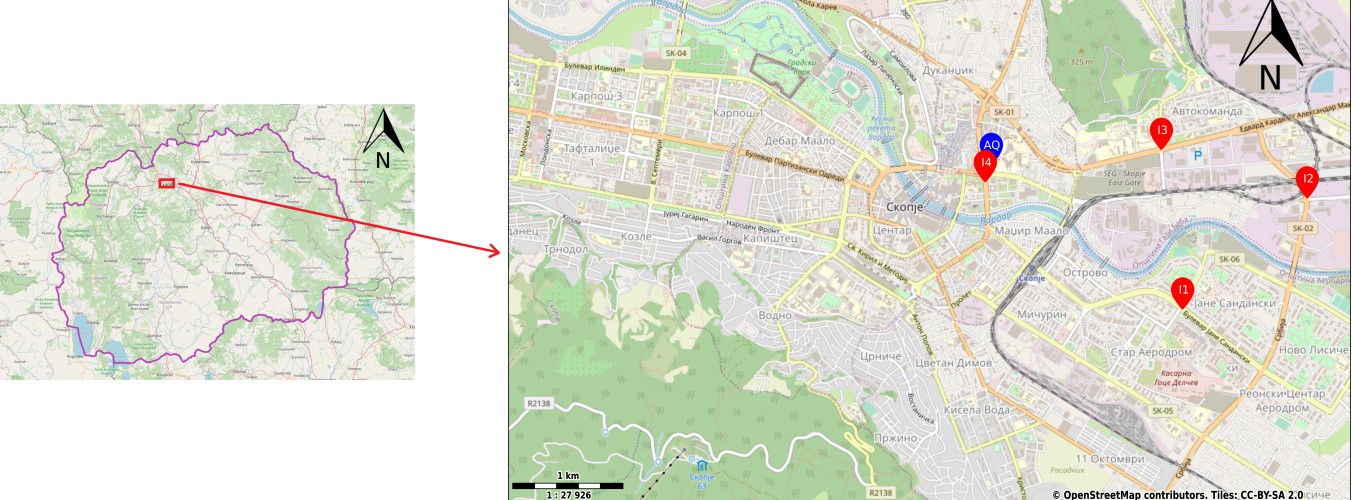

**Figure 1.** Map of North Macedonia (**left**) and the city of Skopje (**right**), with micro-locations of traffic and pollution measurements. Orange roads indicate main city corridors, yellow roads are main streets.

**Table 1.** Traffic-flow dataset summary.

| Intersection | Unit | Number of Measurements | Mean | Min | Max | Std. Dev. |
|---|---|---|---|---|---|---|
| $I_1$ | veh/h | 5535 | 2224.4 | 7 | 5514 | 1524.9 |
| $I_2$ | veh/h | 5015 | 1569.6 | 2 | 3921 | 1023.5 |
| $I_3$ | veh/h | 5483 | 1731.1 | 2 | 3878 | 1280.2 |
| $I_4$ | veh/h | 5588 | 2470.2 | 5 | 6191 | 1622.9 |

**Table 2.** Fuel type and average age of registered vehicles in Skopje per year.

| Fuel Type | Registered Vehicles per Year | | |
|---|---|---|---|
| | **2018** | **2019** | **2020** |
| Gasoline | 92,232 | 87,044 | 87,236 |
| Diesel | 78,836 | 82,591 | 89,513 |
| Gasoline-gas | 2205 | 1938 | 1856 |
| Electricity | 49 | 77 | 92 |
| Alcohol | 1 | 1 | 0 |
| Total | 173,421 | 171,731 | 178,778 |
| Average age [years] | 19.1 | 19.3 | 19.4 |

### 3.1.2. Air-Quality Dataset

The data regarding air-quality measurements were obtained from the Ministry of Environment and Physical Planning of North Macedonia. The data were recorded on a site several meters away from intersection $I_1$, which is one of those featured in the traffic-flow dataset. The dataset consists of measurements of air pollutants: carbon monoxide (CO), nitrogen dioxide ($NO_2$), ozone ($O_3$), and particulate matter of 10 μm ($PM_{10}$). The data were collected at one-hour resolution every day in 2018, 2019, and 2020. The summary statistics of the air-quality dataset are shown in Table 3.

**Table 3.** Air-quality dataset summary.

| Pollutant | Unit | Number of Measurements | Mean | Min | Max | Std. Dev. |
|---|---|---|---|---|---|---|
| CO | mg/m$^3$ | 23,533 | 0.7344 | 0.0116 | 10.5444 | 0.8102 |
| NO$_2$ | μg/m$^3$ | 19,132 | 41.1042 | 0.0191 | 252.8439 | 26.0108 |
| O$_3$ | μg/m$^3$ | 25,786 | 29.9701 | 0.2600 | 149.1398 | 26.7427 |
| PM$_{10}$ | μg/m$^3$ | 23,689 | 60.0136 | 0.2300 | 816.7890 | 58.6379 |

### 3.1.3. Meteorological Dataset

The third dataset used was for the meteorological measurements. The data were obtained from the Copernicus project. The dataset consists of measurements of meteorological conditions in the city of Skopje: wind speed, the temperature at two meters, precipitation, snow depth, and snow cover. The data were collected in one-hour resolution for all days of the years 2018, 2019, and 2020. The summary statistics of the meteorological dataset are shown in Table 4.

**Table 4.** Meteorology dataset summary.

| Measurement | Unit | Number of Measurements | Mean | Min | Max | Std. Dev. |
|---|---|---|---|---|---|---|
| U Wind | m/s | 26,087 | 0.3258 | −3.3958 | 3.5667 | 0.8421 |
| V Wind | m/s | 26,087 | −0.3468 | −5.4112 | 3.9750 | 1.0495 |
| Temperature | °C | 26,087 | 13.6590 | −20.8945 | 38.1836 | 9.4748 |
| Precipitation | m | 26,087 | 0.0869 | 0.0000 | 3.2916 | 0.2487 |
| Snow depth | m | 26,087 | 0.0099 | 0.0000 | 0.2686 | 0.0335 |
| Snow cover | % | 26,087 | 7.5728 | 0.0000 | 100.0000 | 21.3042 |

### 3.1.4. COVID-19 Data and Measures

In addition to the collected data about traffic, air quality, and meteorology, a record was made of all characteristic events regarding the spread of COVID-19 in Skopje during the year 2020; some important global events were also included for clarity. Key dates and measures are presented in Table 5. The most important COVID-19 events in the scope of this paper are the imposed curfews and restrictions on mobility, as those directly impacted the measured traffic flow and emission of air pollutants from vehicles.

**Table 5.** Major COVID-19 events and measures in North Macedonia and the world.

| Date | Event/Measure |
|---|---|
| 30 January 2020 | World Health Organization (WHO) declares COVID-19 a pandemic |
| 26 February 2020 | First reported COVID-19 case in North Macedonia |
| 16 March 2020 | International airports Skopje and Ohrid closed |
| | Borders closed for foreigners |
| 18 March 2020 | State of emergency declared |
| 22 March 2020 | First reported COVID-19 fatality in North Macedonia |
| | Curfew from 9 p.m. to 6 a.m. |
| 8 April 2020 | Curfew from 4 p.m. to 5 a.m. with a complete ban on movement on weekends |
| 22 April 2020 | Curfew from 7 p.m. to 5 a.m. |
| 27 May 2020 | Curfew lifted |
| 8 June 2020 | Curfew from 9 p.m. on Thursday to 5 a.m. on Monday |
| 16 June 2020 | Curfew lifted |

### 3.2. Methodology

To create a linear regression model of air pollution, the methodology consisted of several steps. The first step was to perform data preprocessing and initial observations of data trends, including correlation analysis between all collected variables. Following

the initial observations, a frequency analysis in the frequency domain was used to identify underlying high amplitude frequencies in data. Finally, several regression models are suggested, considering the identified trends in the data. The entire methodology can be summarized with the flowchart shown in Figure 2.

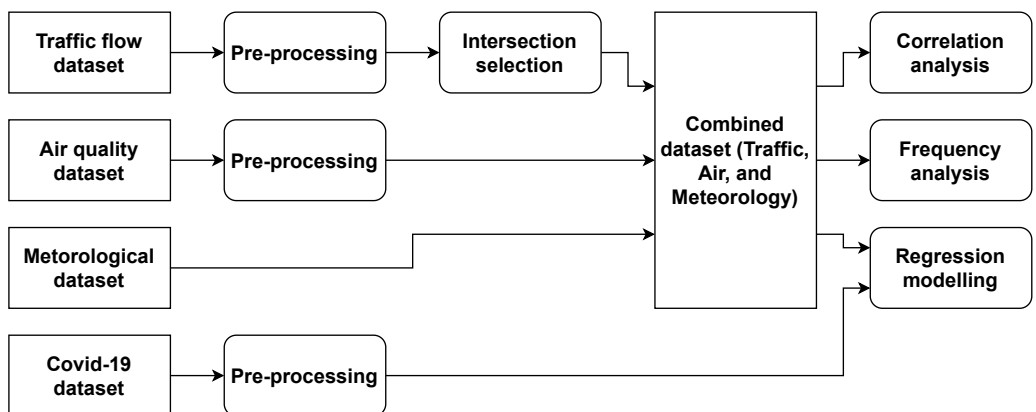

**Figure 2.** Flowchart of the methodology.

### 3.2.1. Initial Observations and Preprocessing

The general behavior of monthly mean traffic flow is shown in Figure 3. The data show the clear impact of the COVID-19 curfews shown in Table 5 on the traffic flow at all four intersections. It is also noted that for intersections $I_1$ and $I_3$, the general monthly behavior of traffic remained the same for all three observed years, excluding the period from April to June during which the curfews were in effect. For intersections $I_2$ and $I_4$, a significant drop in traffic flow was seen during the year 2020. Considering the data in Table 2 showing that the number of registered vehicles was higher in 2020, it is the opinion of the authors that a redistribution of traffic flows occurred between years 2019 and 2020, which caused the drop in mean traffic flow through intersections $I_2$ and $I_4$. The cause for this redistribution could either be due to COVID-19's effects on driving behavior or infrastructure changes. This possible redistribution is beyond the scope of this paper but should be explored in future analysis. Finally, it can be observed that, in general, the traffic flow tends to decrease from July until September during the summer season.

Monthly mean air-quality measurements are shown in Figure 4. From the figure, it can be observed that there are strong seasonal patterns in air-quality measurements. The concentration significantly increases during winter periods for CO and $PM_{10}$. This is probably due to emissions from heating systems. The values of $O_3$ are significantly higher during summer months which is expected since ground-level $O_3$ generation is primarily caused by the decomposition of $NO_2$ and Volatile Organic Components (VOC) under the effects of sunlight. The same effect is the probable reason why $NO_2$ levels are somewhat higher during winter periods. Considering the impact of COVID-19 curfews, the CO and $PM_{10}$ levels are slightly below average during 2020, but since there is a high variance in the data measurements, this effect is hardly visible. The $NO_2$ levels do show a significant drop during curfew periods and a gradual return to previous levels near the end of the year 2020. The concentration of $O_3$ was increased in 2020, which is expected considering the usual inverse correlation with $NO_2$ [17]. The same increase in $O_3$ concentrations in 2020 was observed in [18,19] where the increase in $O_3$ is attributed to lower $O_3$ titration by NO as a consequence of large reductions in $NO_x$. Readers more interested in the interaction of $O_3$ and $NO_2$ are referred to [20].

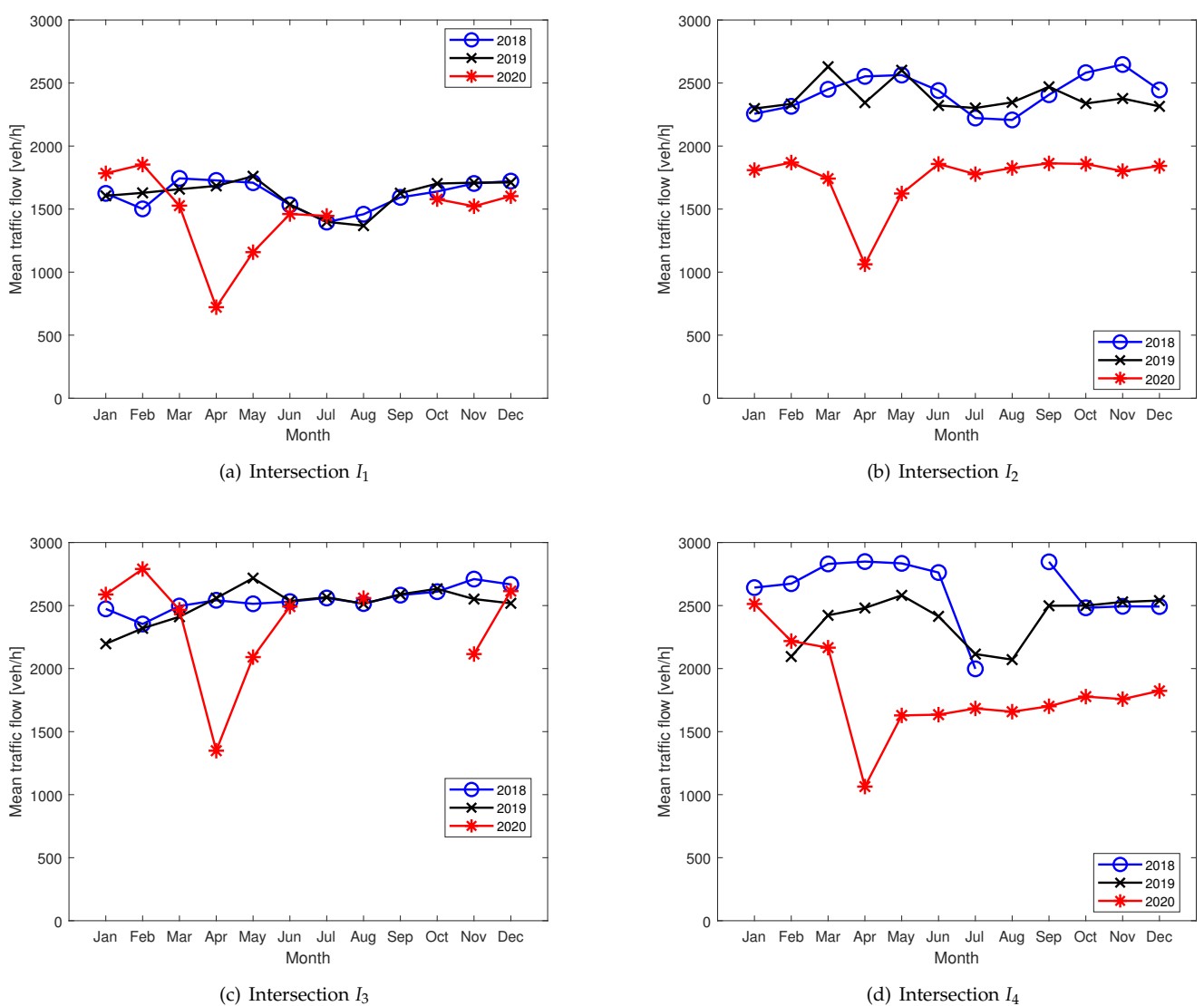

**Figure 3.** Mean monthly traffic flows on intersections for years 2018, 2019 and 2020.

To build a regression model, it is useful to identify correlations between all collected data variables. The correlation matrix of Pearson coefficients is shown in Figure 5. Strong positive correlations were observed between traffic flows at all four intersections, which was expected since intersections are geographically close to each other and capture the same day-to-day human behavior trends. A moderate positive correlation was observed between the $NO_2$ concentration and traffic flows, indicating a possible causal relationship. Concentrations of $PM_{10}$ and CO have a strong correlation, indicating that the sources of their emissions could be the same. Temperature correlates with $O_3$ concentration, which was expected, as $O_3$ levels are higher during the day and during summer. Negative correlations are observed between the Wind U component and traffic flows. Considering the general wind speed in Skopje, it is unlikely that there is any connection between wind direction and traffic flows; thus, this negative correlation is not considered significant. Negative correlations can also be observed between $O_3$ and CO concentrations, which is probably the result of CO having high winter values and $O_3$ having high summer values. The negative correlation between $O_3$ and $NO_2$ concentrations is probably the result of their diurnal patterns and their mutual chemical relationship. No correlation can be observed between CO concentration and traffic flows, which might indicate that there are no CO emissions from traffic. This is probably not the case, and the low correlation values are the results of

high variance in CO measurements and a strong yearly period dynamic. Other variables mostly have very low positive or negative correlations and are not considered significant.

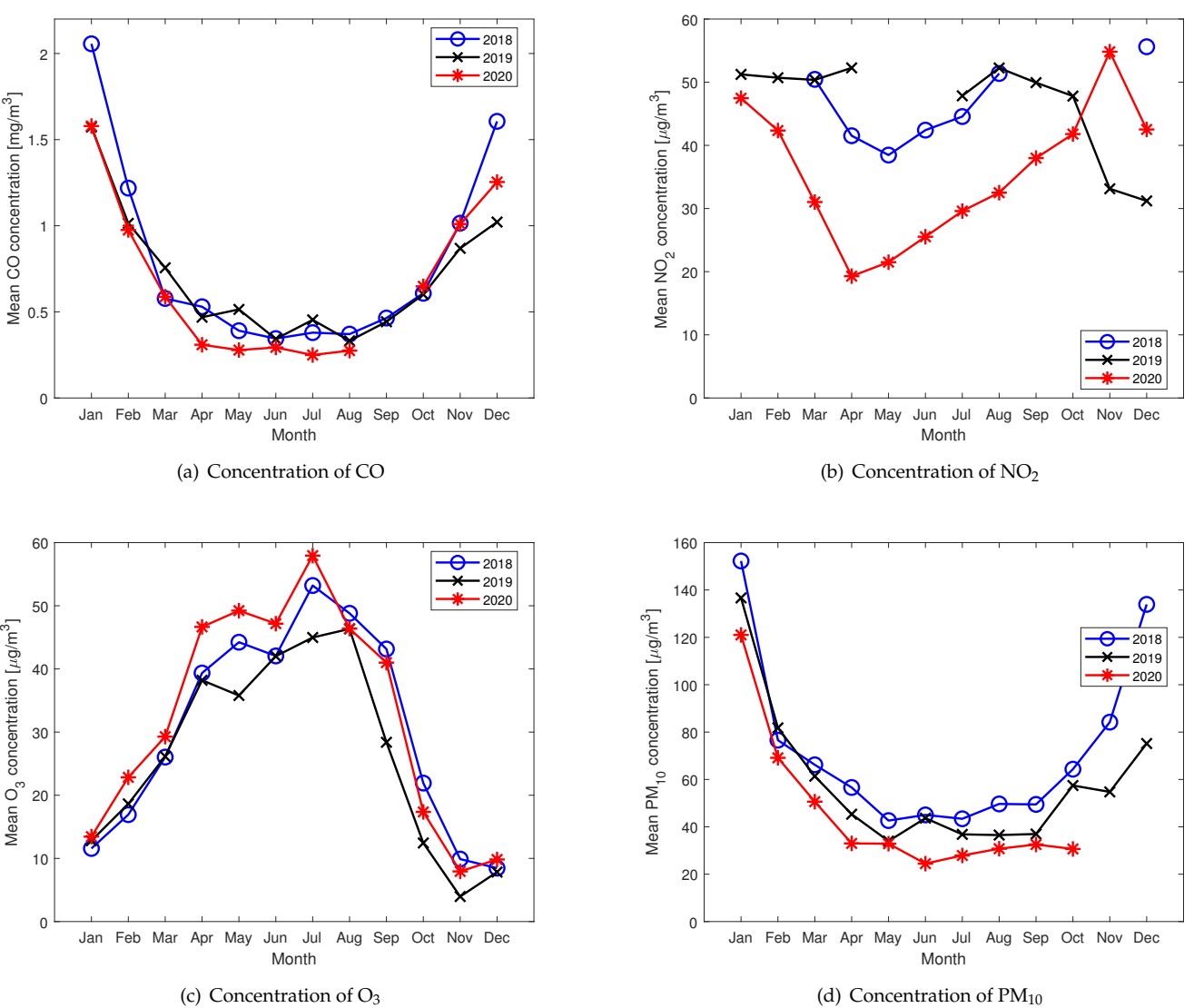

**Figure 4.** Mean monthly concentrations of pollutants for years 2018, 2019 and 2020.

When analyzing any kind of air pollution measurements, it is important to identify if the pollution source is locally emitted or carried by the wind from an external far away location. A useful tool to determine the pollution source direction is the bivariate polar plot of pollutant concentration and wind speed and direction. Such a plot is shown in Figure 6. The plots in Figure 6 are separated into four quarters of the year to separate the seasonal influence. It should be noted that regardless of the quarter, the wind speeds in Skopje are relatively low due to its basin geographical location. Due to this, the effects of local pollution generators are more intense. From the plots of CO and wind, it can be observed that the highest values of CO occur in winter periods, particularly when the wind is blowing from north to east or south–east. The former direction corresponds to the location of a steel processing plant (2.8 km distance), and the latter corresponds to the location of a thermal power plant (1.2 km distance). For $NO_2$, there are no visible patterns present indicating local emissions. For $O_3$, it is observed that at wind very low wind speeds, the concentration has lower values. This can indicate that either most observed $O_3$ is produced elsewhere or that $O_3$ production is hindered by a local presence of $NO_2$. Some rare occurrences of high $O_3$ levels could also be the result of stratospheric $O_3$ breaches.

For PM$_{10}$, the pattern points to local emissions with a slight influence of north–east and south–east winds, similarly to CO observations.

**Figure 5.** Pearson correlation matrix of all data variables.

### 3.2.2. Frequency Analysis

Since the collected data are in the form of a time series, frequency analysis can be used to identify the short and long-term fluctuations in the data. The short-term fluctuations usually depend on local phenomena such as traffic emissions, and the long-term fluctuations correspond to seasonal variations. The time-series data need to be complete to perform the frequency analysis. Hence, the missing values are filled with the nearest not missing value. Missing values were not filled using interpolation, as there are several wide gaps in the data, which could possibly introduce false signals in the data. To perform the analysis, a time series $X_t$ with length $N$ can be represented as a linear combination of harmonic functions with frequencies $f_j$ and amplitudes $A_j$ and $B_j$, as shown in Equation (1).

$$X_t = \mu + \sum_{j=1}^{N/2} \left[ A_j cos(2\pi f_j t) + B_j sin(2\pi f_j t) \right],$$

$$t = 1, 2, \ldots, N,$$

(1)

where $\mu$ is a constant, $N$ is the length of the time series $X_t$, and $f_j$ are frequencies related to the $N$ according to:

$$f_j \equiv j/N, 1 \le j \le N/2.$$

(2)

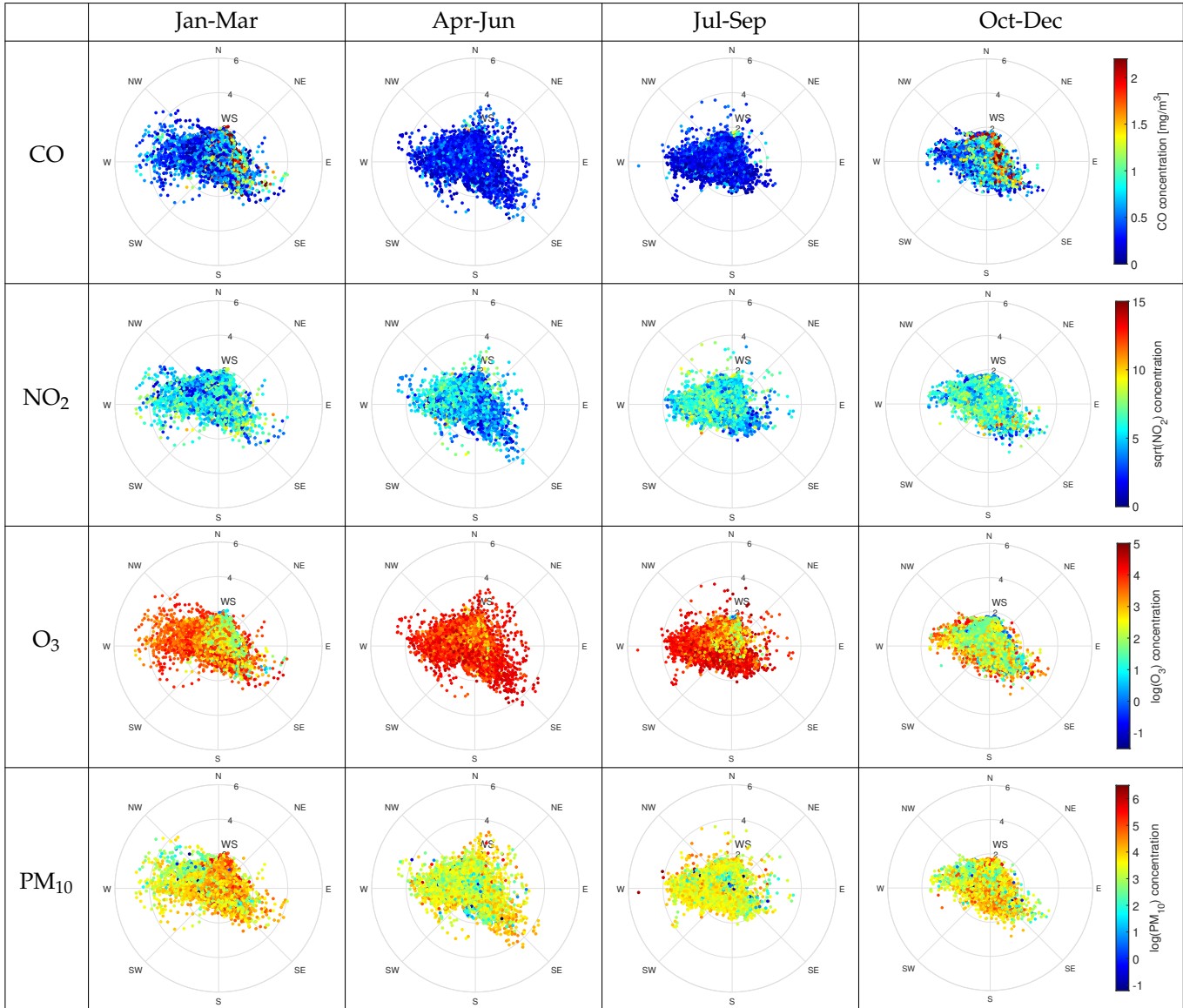

**Figure 6.** Bivariate polar plots for hourly mean pollutants' concentration levels. CO concentration in μg/m³; $NO_2$ concentration shown as $sqrt(NO_2)$; $O_3$ concentration shown as $log(O_3)$; $PM_{10}$ concentration shown as $log(PM_{10})$; WS—wind speed (m/s).

Since all collected data are sampled hourly, the highest frequency that can be analyzed is a period of 2 h ($f = 0.5$ 1/h) according to the Nyquist–Shannon sampling theorem, which states that the data sample rate must be at least twice the highest frequency to accurately reproduce the signal. To separate the time-series data into a linear combination of harmonic functions, the fast Fourier transform (FFT) algorithm was used, and the results are shown in the form of a periodogram in Figure 7 with power spectral densities (PSDs) for pollution data and Figure 8 for traffic data. To better identify key frequencies, the periodograms were smoothed using Gaussian smoothing with a window size of 200 data samples to reduce the effect of noise. In pollution data periodograms, strong amplitudes were detected for frequencies with periods of 24, 12, 8, and 6 hours. High amplitudes were also observed in the low-frequency range, which corresponds to a period of one year. Small peaks were observed in higher frequencies but are not considered significant. In traffic data periodograms, strong amplitudes were detected for the same frequencies as in the pollution data. However, higher frequencies in traffic data show significant oscillations for higher harmonic periods of 4.8, 4, and 3.43 h. The identified peaks show harmonic properties, which means that a fundamental frequency or the lowest harmonic can be identified with a

period of 24 h. Since the time series of both pollution and traffic data are composed of a fundamental frequency with a period of 24 h and corresponding harmonic frequencies, the phase shift of each frequency can be calculated to identify if there is a time delay between them. The calculated phase shifts of fundamental frequencies are shown in Table 6. The phase shift calculated in radians can be translated to a time shift for easier interpretation of how the signals are delayed from one another. From the table, it can be observed that intersection data have similar phase shifts, which was expected considering the correlation analysis in the previous section. The phase shift of $O_3$ data are also similar to intersection data, which explains the high correlation observed earlier. The phase shift of CO shows that in regard to intersection data, it was delayed by six to eight hours. A similar delay was observed between $PM_{10}$ and intersection data. The phase shift of $NO_2$ shows that it is delayed by two to four hours in regard to intersection data.

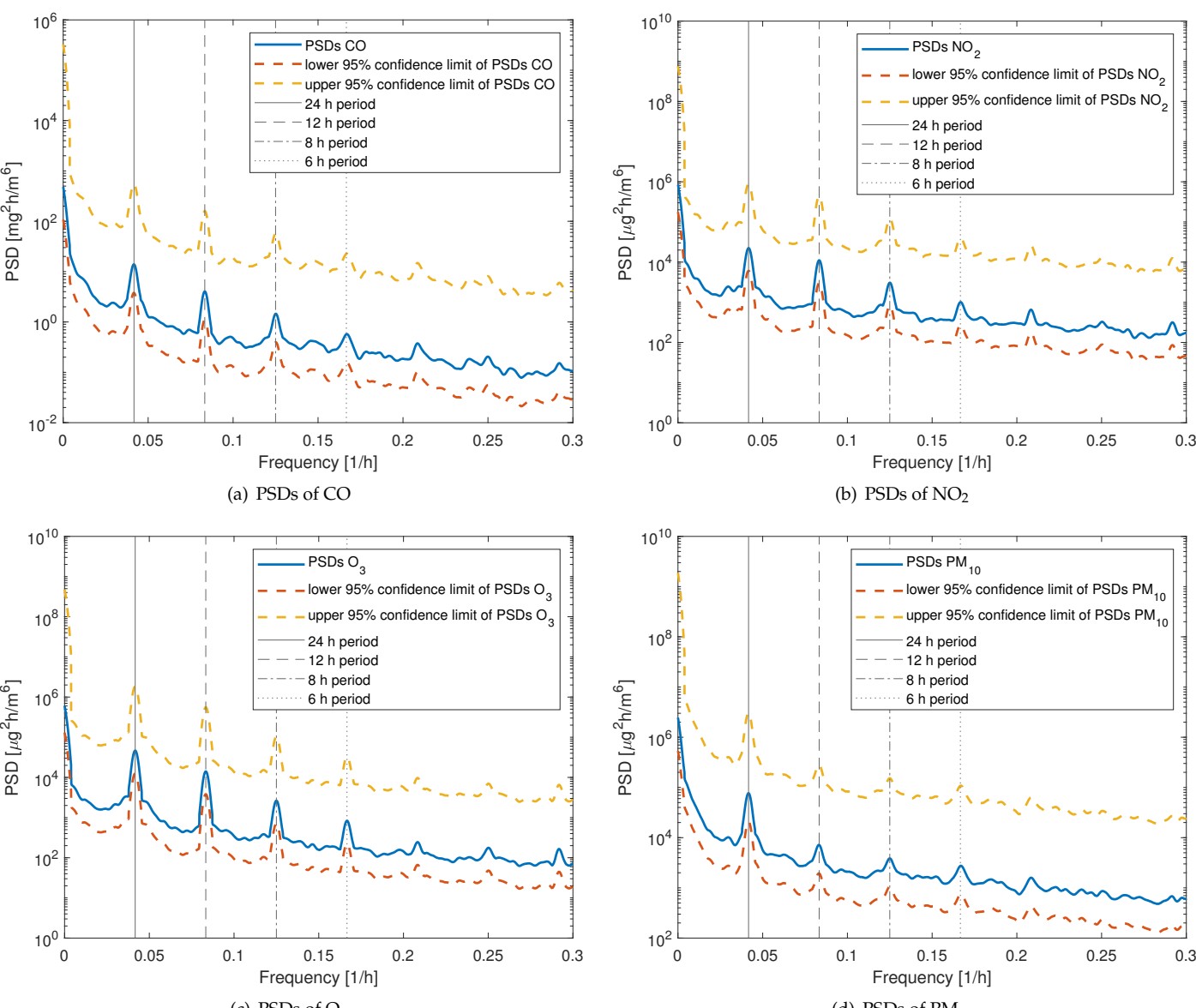

**Figure 7.** Power spectral densities (PSDs) for hourly pollution concentration measurements.

**Table 6.** Phases and time shifts of time-series data at the fundamental frequency with a period of 24 h.

| Time Series | Phase Shift [rad] | Time Shift [h] |
| :---: | :---: | :---: |
| $I_1$ | 2.2678 | 8.6622 |
| $I_2$ | 2.6356 | 10.0674 |
| $I_3$ | 2.6238 | 10.0221 |
| $I_4$ | 2.4597 | 9.3953 |
| CO | 0.6291 | 2.4031 |
| $NO_2$ | 1.7058 | 6.5159 |
| $O_3$ | 2.5671 | 9.8057 |
| $PM_{10}$ | 0.7897 | 3.0165 |

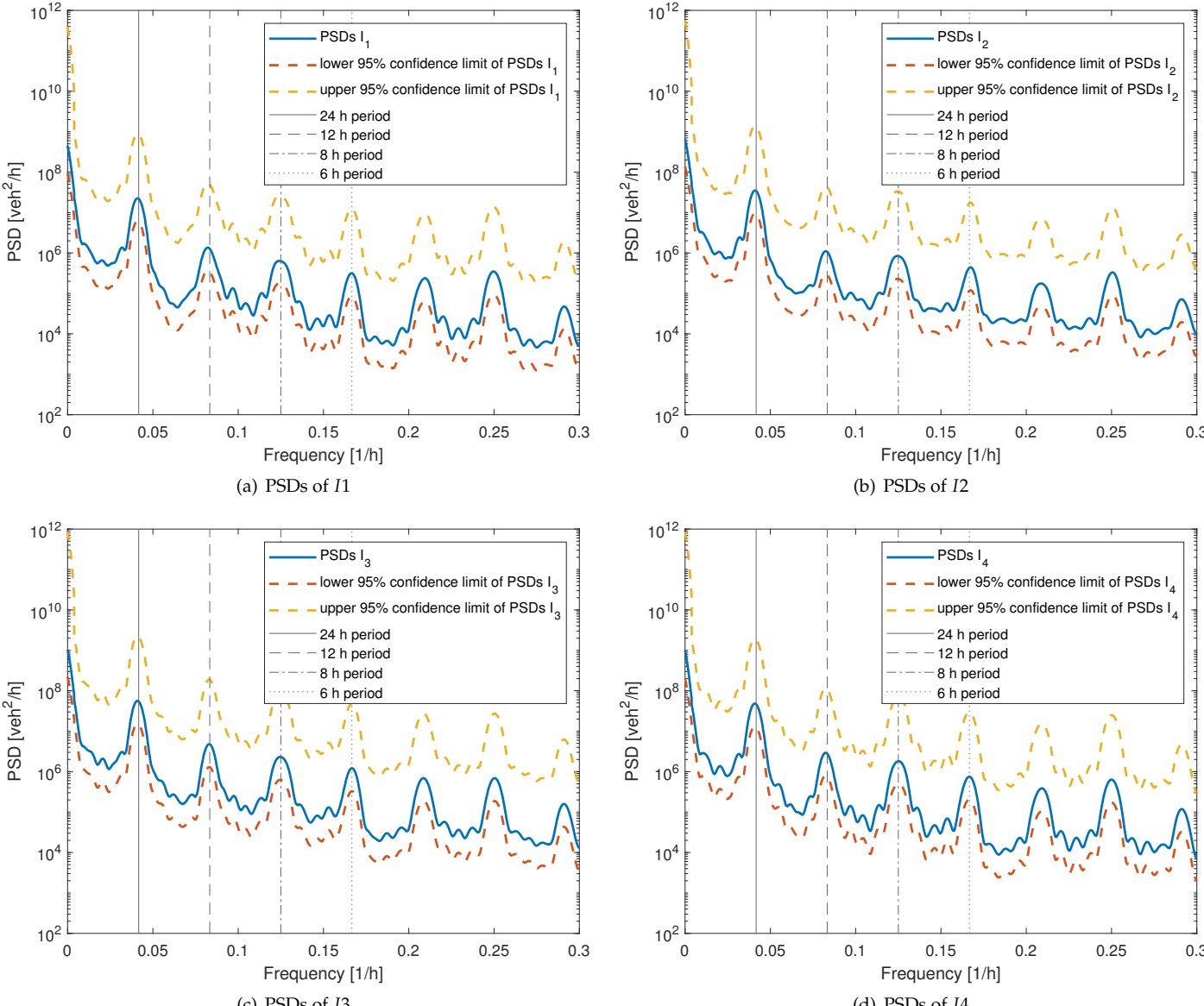

(a) PSDs of *I*1

(b) PSDs of *I*2

(c) PSDs of *I*3

(d) PSDs of *I*4

**Figure 8.** Power spectral densities (PSDs) for hourly traffic flow measurements.

### 3.3. Regression Modeling

Considering the literature review and the initial observations of the data from the previous section, it should be possible to construct a linear regression model to model air pollution. Considering COVID-19 restrictions' impact on traffic flow and possibly air pollution, three separate groups of regression models were analyzed. The first group only

used the data from the years 2018 and 2019. The second group used only the data from 2020, including the periods with COVID-19 restrictions. The third group used all available data. By comparing the obtained regression model coefficient from the three groups, the impacts of COVID-19 measures can be analyzed appropriately. The rationale behind this approach is that values measured during COVID-19 measures would usually fall into an outlier category and might be overlooked by traditional analysis. In all models, the dependent variable was one of the observed pollutants. The independent variables was traffic and meteorological data or their combinations. The impact and significance of independent variables was analyzed with a t-test with a significance level of 95%.

## 4. Results

In this section, the results of several proposed linear regression models are presented with a clear distinction of how the regression coefficients change depending on the three identified data groups.

### 4.1. Modeling of CO Pollution

Taking into account the high variance of CO data and possible resultant heteroscedasticity, the CO data were log transformed to reduce the variance of residuals in the regression model. Since there is a high correlation between traffic flows at all identified intersections, only the data from one intersection can be included in the model. For this reason, the data from each intersection were combined using the mean average to obtain the general behavior of the traffic flow. The results of the proposed linear regression models of ln(CO) are presented in Table 7. The calculated regression coefficients and associated metrics of the ln(CO) regression model are presented in detail for each model in Tables A1–A6 in the Appendix A of this paper.

**Table 7.** Comparison of ln(CO) linear regression models for hourly data.

| Model | 2018–2019 | 2020 | 2018–2020 | AR | DF | $R^2$ | Adjusted $R^2$ | RMSE | F-Stat | *p*-Value |
|-------|-----------|------|-----------|-----|------|-------|----------------|------|--------|-----------|
| Model 1 | ✓ | n.a. | n.a. | n.a. | 3957 | 0.271 | 0.270 | 0.708 | 294 | 0.00000 |
| Model 2 | n.a. | ✓ | n.a. | n.a. | 1553 | 0.447 | 0.445 | 0.707 | 251 | 0.00000 |
| Model 3 | n.a. | n.a. | ✓ | n.a. | 5515 | 0.312 | 0.311 | 0.724 | 499 | 0.00000 |
| Model 4 | ✓ | n.a. | n.a. | ✓ | 3943 | 0.751 | 0.750 | 0.412 | 1980 | 0.00000 |
| Model 5 | n.a. | ✓ | n.a. | ✓ | 1549 | 0.816 | 0.816 | 0.408 | 1148 | 0.00000 |
| Model 6 | n.a. | n.a. | ✓ | ✓ | 5499 | 0.776 | 0.776 | 0.412 | 3171 | 0.00000 |

### 4.2. Modeling of $NO_2$ Pollution

The $NO_2$ data were transformed to sqrt($NO_2$) before regression modeling to prevent heteroscedasticity in the data. Square root transformation was used instead of log transformation, since log transformation was found to be too aggressive for $NO_2$ data. The same independent variables were used to model sqrt($NO_2$) as in the ln(CO) model. The results of the proposed linear regression models of sqrt(CO) are presented in Table 8. The calculated regression coefficients and associated metrics of the sqrt($NO_2$) regression model are presented in detail for each model in Tables A7–A12 in the Appendix A of this paper.

### 4.3. Modeling of $PM_{10}$ Pollution

The $PM_{10}$ data were transformed to $log(PM_{10})$ before regression modeling to prevent heteroscedasticity in the data. The same independent variables were used to model ln($PM_{10}$) as in the ln(CO) and sqrt($NO_2$) model. The results of the proposed linear regression models of $sqrt(CO)$ are presented in Table 9. The calculated regression coefficients of the $log(PM_{10})$ regression model and associated metrics are presented in detail for each model in Tables A13–A18 in the Appendix A of this paper.

**Table 8.** Comparison of sqrt($NO_2$) linear regression models for hourly data.

| Model | 2018–2019 | 2020 | 2018–2020 | AR | DF | $R^2$ | Adjusted $R^2$ | RMSE | F-Stat | *p*-Value |
|---|---|---|---|---|---|---|---|---|---|---|
| Model 7 | ✓ | n.a. | n.a. | n.a. | 2486 | 0.152 | 0.150 | 1.89 | 89 | 0.00000 |
| Model 8 | n.a. | ✓ | n.a. | n.a. | 1925 | 0.327 | 0.325 | 1.60 | 187 | 0.00000 |
| Model 9 | n.a. | n.a. | ✓ | n.a. | 4417 | 0.223 | 0.222 | 1.82 | 253 | 0.00000 |
| Model 10 | ✓ | n.a. | n.a. | ✓ | 2484 | 0.604 | 0.603 | 1.29 | 630 | 0.00000 |
| Model 11 | n.a. | ✓ | n.a. | ✓ | 1922 | 0.741 | 0.740 | 0.99 | 916 | 0.00000 |
| Model 12 | n.a. | n.a. | ✓ | ✓ | 4413 | 0.677 | 0.676 | 1.18 | 1540 | 0.00000 |

**Table 9.** Comparison of ln($PM_{10}$) linear regression models for hourly data.

| Model | 2018–2019 | 2020 | 2018–2020 | AR | DF | $R^2$ | Adjusted $R^2$ | RMSE | F-Stat | *p*-Value |
|---|---|---|---|---|---|---|---|---|---|---|
| Model 13 | ✓ | n.a. | n.a. | n.a. | 3976 | 0.151 | 0.150 | 0.762 | 141 | 0.00000 |
| Model 14 | n.a. | ✓ | n.a. | n.a. | 1352 | 0.221 | 0.218 | 0.803 | 77 | 0.00000 |
| Model 15 | n.a. | n.a. | ✓ | n.a. | 5334 | 0.180 | 0.179 | 0.780 | 235 | 0.00000 |
| Model 16 | ✓ | n.a. | n.a. | ✓ | 3973 | 0.664 | 0.664 | 0.480 | 1310 | 0.00000 |
| Model 17 | n.a. | ✓ | n.a. | ✓ | 1358 | 0.595 | 0.593 | 0.579 | 330 | 0.00000 |
| Model 18 | n.a. | n.a. | ✓ | ✓ | 5331 | 0.653 | 0.652 | 0.508 | 1670 | 0.00000 |

## 5. Discussion

Results of the ln(CO) regression models given in Table 7 show that all models were significant, having *p*-values well below 0.05. Since time-series data such as pollution measurements are usually subject to the autoregression of residuals, it is difficult to trust the obtained *p*-values. Models 4–6 included the AR component, resulting in much higher $R^2$ values. In those models, the residuals were no longer autocorrelated, and instead followed a normal distribution. Hence, the models are considered significant. By looking at the obtained regression coefficients and associated *p*-values of each component, it can be observed that the precipitation component was not significant in all models except model 2, where it barely passed the significance test. All other components were confirmed to be significant, except the U wind component in model 4. It was observed that the coefficient of traffic was positive and significant in all models. This observation suggests that traffic does have an impact on the CO concentration. However, air temperature has the strongest influence on CO concentration, which is to be expected, considering the strong negative correlation between air temperature and CO concentration. The interpretation of this result nevertheless remains difficult, as the effect of temperature on CO concentration is not direct. It is the opinion of the authors that for the given case study, the following factors connect the air temperature and CO concentration:

- Heating systems' activation during colder periods;
- Formation of inversion layers above the urban areas during colder periods;
- Idling of vehicles with the engine running for defrosting and heating.

Since the factors mentioned above are difficult to measure, and their negative correlation with temperature was expected, the authors deem the inclusion of temperature in the regression model appropriate. It is observed that the $R^2$ value is somewhat higher for models with data only from 2020, but the regression coefficients differ only slightly. Since the coefficients are similar in all models, it is the opinion of the authors that periods with restrictive COVID-19 measures did not create an outlier.

Considering the results of the sqrt($NO_2$) regression models shown in Table 8, similar conclusions in regard to autoregression of residuals can be made as in ln(CO) models. The primary concern with the sqrt($NO_2$) models is a large amount of missing data, especially in years 2018–2019, as can be seen in Figure 4b. The *p*-values of the sqrt($NO_2$) model components show that precipitation does not seem to be a significant factor in the model.

For models 7 and 10, the *p*-values of coefficients show that air temperature and precipitation are not significant. In all sqrt($NO_2$) models, the traffic component is significant and positive. The impact of COVID-19 measures on $NO_2$ pollution seems evident, as a large drop in $NO_2$ levels as found for the same time as most restrictive curfews were imposed. The obtained regression models also support this, as no outliers were identified in periods with restrictive COVID-19 measures. However, a direct comparison between the years 2020 and 2018–2019 was difficult, due to missing data.

In the correlation analysis, there was a high positive correlation between CO and $PM_{10}$ concentrations. This correlation is high, since both CO and $PM_{10}$ share strong yearly periodicity. This high correlation might imply that similar regression models could be used to model both CO and $PM_{10}$ concentrations. From the results of $\ln(PM_{10})$ regression models in Table 9, it is evident that models' performance levels considering $R^2$ were lower than in the case of $\ln(CO)$ models. The traffic component was significant and positive in all $\ln(PM_{10})$ models. Unlike the $\ln(CO)$ and sqrt($NO_2$) regression models, the precipitation component remained significant in all $\ln(PM_{10})$ models. The negative coefficient of precipitation implies that the presence of rain decreases the concentration of $PM_{10}$ particles in the air. The impact of COVID-19 measures is not significant in regard to $PM_{10}$ concentration. This is attributed to the fact that many external sources of $PM_{10}$ other than vehicle emissions are located in the vicinity of Skopje.

The results of COVID-19's influence on air pollution mostly comply with other research on this topic summarized in [16]. In most case studies, the immediate effect was observed in the reductions in $NO_2$ and *PM* levels and a smaller increase in $O_3$. In this research the effect was only noticed for $NO_2$ and $O_3$, and $PM_{10}$ levels showed only a small decrease.

## 6. Conclusions

In this paper, air pollution in urban areas was modeled using linear regression. The research was conducted as a case study in the city of Skopje, North Macedonia. Collected traffic, air quality, and meteorological data were used to create a regression model of $\ln(CO)$, sqrt($NO_2$), and $\ln(PM_{10})$ concentrations. The traffic component was found to be significant in all regression models with a positive coefficient. However, the impact of traffic on air pollution was strong only in terms of $NO_2$ concentration and low for CO and $PM_{10}$ due to strong seasonal influence. Additionally, the impact of COVID-19-related restrictive measures was analyzed. The results show that a significant drop in $NO_2$ concentration coincided with the imposed curfews and bans on human mobility. The values of CO and $PM_{10}$ did not change significantly during COVID-19 restrictions, indicating that traffic is not the primary source of CO and $PM_{10}$ emissions. Finally, to answer the main research question, it can be concluded that COVID-19-related restrictive measures did not create an outlier in the regression model and that the regression coefficients mostly remained the same. From the results, several policy recommendations arise for the identified stakeholders. For governing officials, the primary concern should be high levels of air pollution, especially in winter periods. Incentives should be made to reduce unnecessary road traffic and to switch to newer vehicles, which have reduced emissions. However, it is also evident that a large contributor to pollution is the industry located in the city. In the short term, modernization of industry could provide large benefits to reduced air pollution. In the long term, the industrial sector should be separated from the city to a less-populated area. The primary limitation of this research was the resolution of the observed data variables, and there were many missing values, particularly for intersection data, since only a week of data were available for each month. Adding more air-quality-measurement stations throughout the city would greatly improve the results and allow for spatial analysis of data. The regression models could be improved by including more variables, such as air pressure, as it could explain some of the variability in air pollution data. The COVID-19 measures provided a unique opportunity to observe how the sudden reduction of traffic might affect air quality. For the city of Skopje, however, the primary pollutants are not the vehicles, but instead, possibly the steel processing plant and power plant located in the city.

Future work on this topic should include the collected traffic and pollution data for years following the COVID-19 pandemic.

**Author Contributions:** The conceptualization of the study was conducted by M.M., B.A., T.F. and E.I. The funding acquisition was conducted by E.I. The writing of the original draft and preparation of the paper was conducted by M.M. All authors contributed to the writing of the paper and final editing. The supervision was conducted by B.A., T.F. and E.I. Visualizations were conducted by M.M. All authors have read and agreed to the published version of the manuscript.

**Funding:** This research was funded by the Croatian Science Foundation under the project UIP-2019-04-1737, and the project IP-2020-02-5042, and by the European Regional Development Fund under the grant KK.01.1.1.01.0009 (DATACROSS).

**Institutional Review Board Statement:** Not applicable.

**Informed Consent Statement:** Not applicable.

**Data Availability Statement:** The data presented in this study are available on request from the corresponding author. The data regarding air-quality measurements are publicly available from the Ministry of Environment and Physical Planning of North Macedonia and can be found here: https://air.moepp.gov.mk/ (Accessed on 20 November 2022). The data regarding meteorological measurements are publicly available and can be found here: https://cds.climate.copernicus.eu. (Accessed on 20 November 2022). The data regarding traffic flow measurements were obtained from the Centre for Traffic Management and Control (CUKS)—Skopje, and are available from them upon request.

**Acknowledgments:** This research has also been carried out within the activities of the Centre of Research Excellence for Data Science and Cooperative Systems supported by the Ministry of Science and Education of the Republic of Croatia.

**Conflicts of Interest:** The authors declare no conflict of interest. The funders had no role in the design of the study; in the collection, analyses, or interpretation of data; in the writing of the manuscript; or in the decision to publish the results.

## Abbreviations

The following abbreviations are used in this manuscript:

| | |
|---|---|
| SARS-CoV-2 | Severe acute respiratory syndrome coronavirus 2 |
| COVID-19 | Coronavirus disease of 2019 |
| CO | Carbon monoxide |
| $NO_2$ | Nitrogen dioxide |
| $O_3$ | Ozone |
| $PM_{2.5}$ | Particulate matter of 2.5 [μm] |
| $PM_{10}$ | Particulate matter of 10 [μm] |
| WHO | World Health Organization |
| VOC | Volatile organic compounds |
| PSDs | Power spectral densities |
| FFT | Fast Fourier transform |

## Appendix A

The detailed tables of analyzed regression models for $\ln(CO)$, $\mathrm{sqrt}(NO_2)$ and $\ln(PM_{10})$ are included in this appendix.

The Tables A1–A3 show the calculated regression coefficients of the $\ln(CO)$ linear regression models. The associated standard error (SE), t-statistic, and *p*-values are also shown in the tables for each model component. Tables A4–A6 show the same regression models with the autoregression (AR) $\ln(CO(t-1))$ component included as part of the model.

**Table A1.** Model 1: ln(CO) linear regression model hourly data 2018–2019.

|  | Coefficients | SE | t-Stat | *p*-Value |
|---|---|---|---|---|
| Intercept | −0.44777 | 0.03188 | −14.04706 | 0.00000 |
| Traffic | 0.00020 | 0.00001 | 18.55800 | 0.00000 |
| Temperature | −0.04124 | 0.00126 | −32.85634 | 0.00000 |
| Precipitation | −0.05794 | 0.05469 | −1.05947 | 0.28945 |
| U Wind | 0.08343 | 0.01646 | 5.06833 | 0.00000 |
| V Wind | 0.13757 | 0.01149 | 11.97037 | 0.00000 |

**Table A2.** Model 2: ln(CO) linear regression model hourly data 2020.

|  | Coefficients | SE | t-Stat | *p*-Value |
|---|---|---|---|---|
| Intercept | −0.47425 | 0.04524 | −10.48372 | 0.00000 |
| Traffic | 0.00021 | 0.00001 | 14.20153 | 0.00000 |
| Temperature | −0.05993 | 0.00203 | −29.51000 | 0.00000 |
| Precipitation | −0.23848 | 0.11479 | −2.07749 | 0.03792 |
| U Wind | 0.24775 | 0.02598 | 9.53464 | 0.00000 |
| V Wind | 0.23902 | 0.01937 | 12.34090 | 0.00000 |

**Table A3.** Model 3: ln(CO) linear regression model hourly data 2018–2020.

|  | Coefficients | SE | t-Stat | *p*-Value |
|---|---|---|---|---|
| Intercept | −0.47948 | 0.02642 | −18.14930 | 0.00000 |
| Traffic | 0.00022 | 0.00001 | 24.32205 | 0.00000 |
| Temperature | −0.04653 | 0.00109 | −42.86097 | 0.00000 |
| Precipitation | −0.08443 | 0.04990 | −1.69188 | 0.09073 |
| U Wind | 0.13100 | 0.01414 | 9.26285 | 0.00000 |
| V Wind | 0.16150 | 0.01005 | 16.06434 | 0.00000 |

**Table A4.** Model 4: ln(CO) linear regression model hourly data 2018–2019 with auto-regression.

|  | Coefficients | SE | t-Stat | *p*-Value |
|---|---|---|---|---|
| Intercept | −0.09968 | 0.01900 | −5.24732 | 0.00000 |
| Traffic | 0.00005 | 0.00001 | 8.32368 | 0.00000 |
| Temperature | −0.00917 | 0.00082 | −11.15764 | 0.00000 |
| Precipitation | −0.06108 | 0.03187 | 1.91678 | 0.05534 |
| U Wind | 0.00987 | 0.00963 | 1.02441 | 0.30571 |
| V Wind | 0.02569 | 0.00681 | 3.77092 | 0.00017 |
| $\ln(CO(t-1))$ | 0.80217 | 0.00927 | 86.53954 | 0.00000 |

**Table A5.** Model 5: ln(CO) linear regression model hourly data 2020 with auto-regression.

|  | Coefficients | SE | t-Stat | *p*-Value |
|---|---|---|---|---|
| Intercept | −0.08127 | 0.02703 | −3.00702 | 0.00268 |
| Traffic | 0.00004 | 0.00001 | 4.62462 | 0.00000 |
| Temperature | −0.01252 | 0.00145 | −8.64623 | 0.00000 |
| Precipitation | −0.05001 | 0.06653 | −0.75170 | 0.45235 |
| U Wind | 0.05686 | 0.01537 | 3.69941 | 0.00022 |
| V Wind | 0.06004 | 0.01163 | 5.16384 | 0.00000 |
| $\ln(CO(t-1))$ | 0.80472 | 0.01441 | 55.82959 | 0.00000 |

**Table A6.** Model 6: ln(CO) linear regression model hourly data 2018–2020 with auto-regression.

|  | Coefficients | SE | t-Stat | *p*-Value |
|---|---|---|---|---|
| Intercept | −0.09354 | 0.01548 | −6.04178 | 0.00000 |
| Traffic | 0.00005 | 0.00001 | 9.59341 | 0.00000 |
| Temperature | −0.00962 | 0.00071 | −13.52489 | 0.00000 |
| Precipitation | −0.05537 | 0.02846 | −1.94547 | 0.05177 |
| U Wind | 0.02168 | 0.00813 | 2.66720 | 0.00767 |
| V Wind | 0.03203 | 0.00585 | 5.47261 | 0.00000 |
| ln(CO($t-1$)) | 0.81270 | 0.00766 | 106.14666 | 0.00000 |

The Tables A7–A9 show the calculated regression coefficients of the sqrt($NO_2$) linear regression models. The associated standard error (SE), t-statistic, and *p*-values are also shown in the tables for each model component. Tables A10–A12 show the same regression models with the autoregression (AR) $sqrt(NO_2(t-1))$ component included as part of the model.

**Table A7.** Model 7: sqrt($NO_2$) linear regression model hourly data 2018–2019.

|  | Coefficients | SE | t-Stat | *p*-Value |
|---|---|---|---|---|
| Intercept | 5.00079 | 0.10844 | 46.11649 | 0.00000 |
| Traffic | 0.00073 | 0.00004 | 20.46105 | 0.00000 |
| Temperature | −0.00340 | 0.00414 | −0.82090 | 0.41178 |
| Precipitation | 0.20843 | 0.21188 | 0.98373 | 0.32534 |
| U Wind | 0.40763 | 0.05534 | 7.36536 | 0.00000 |
| V Wind | 0.07784 | 0.03706 | 2.10062 | 0.03577 |

**Table A8.** Model 8: sqrt($NO_2$) linear regression model hourly data 2020.

|  | Coefficients | SE | t-Stat | *p*-Value |
|---|---|---|---|---|
| Intercept | 4.64907 | 0.09311 | 49.92990 | 0.00000 |
| Traffic | 0.00088 | 0.00003 | 28.94455 | 0.00000 |
| Temperature | −0.05411 | 0.00431 | −12.56638 | 0.00000 |
| Precipitation | 0.15871 | 0.12101 | 1.31151 | 0.18984 |
| U Wind | 0.39526 | 0.05267 | 7.50431 | 0.00000 |
| V Wind | 0.20393 | 0.03827 | 5.32891 | 0.00000 |

**Table A9.** Model 9: sqrt($NO_2$) linear regression model hourly data 2018–2020.

|  | Coefficients | SE | t-Stat | *p*-Value |
|---|---|---|---|---|
| Intercept | 4.69737 | 0.07298 | 64.36915 | 0.00000 |
| Traffic | 0.00085 | 0.00002 | 35.06696 | 0.00000 |
| Temperature | −0.02392 | 0.00303 | −7.88787 | 0.00000 |
| Precipitation | 0.04106 | 0.11304 | 0.36320 | 0.71647 |
| U Wind | 0.44506 | 0.03961 | 11.23600 | 0.00000 |
| V Wind | 0.12379 | 0.02742 | 4.51490 | 0.00001 |

**Table A10.** Model 10: sqrt($NO_2$) linear regression model hourly data 2018–2019 with auto-regression.

|  | Coefficients | SE | t-Stat | *p*-Value |
|---|---|---|---|---|
| Intercept | 1.34556 | 0.10113 | 13.30503 | 0.00000 |
| Traffic | 0.00026 | 0.00003 | 10.18534 | 0.00000 |
| Temperature | −0.00363 | 0.00283 | −1.28077 | 0.20039 |
| Precipitation | −0.23729 | 0.14512 | −1.63515 | 0.10214 |
| U Wind | 0.07690 | 0.03841 | 2.00185 | 0.04541 |
| V Wind | −0.00136 | 0.02539 | −0.05374 | 0.95715 |
| sqrt($NO_2$($t-1$)) | 0.71777 | 0.01349 | 53.22452 | 0.00000 |

**Table A11.** Model 11: sqrt($NO_2$) linear regression model hourly data 2020 with auto-regression.

|  | Coefficients | SE | t-Stat | *p*-Value |
|---|---|---|---|---|
| Intercept | 1.03517 | 0.08720 | 11.87149 | 0.00000 |
| Traffic | 0.00025 | 0.00002 | 11.47203 | 0.00000 |
| Temperature | −0.01395 | 0.00277 | −5.03585 | 0.00000 |
| Precipitation | 0.01771 | 0.07515 | 0.23560 | 0.81377 |
| U Wind | 0.10412 | 0.03311 | 3.14472 | 0.00169 |
| V Wind | 0.07518 | 0.02387 | 3.14919 | 0.00166 |
| sqrt($NO_2(t-1)$) | 0.76460 | 0.01380 | 55.42396 | 0.00000 |

**Table A12.** Model 12: sqrt($NO_2$) linear regression model hourly data 2018–2020 with auto-regression.

|  | Coefficients | SE | t-Stat | *p*-Value |
|---|---|---|---|---|
| Intercept | 1.10413 | 0.06560 | 16.83171 | 0.00000 |
| Traffic | 0.00027 | 0.00002 | 15.34896 | 0.00000 |
| Temperature | −0.00729 | 0.00197 | −3.70610 | 0.00021 |
| Precipitation | −0.08839 | 0.07292 | −1.21213 | 0.22553 |
| U Wind | 0.09584 | 0.02595 | 3.69294 | 0.00022 |
| V Wind | 0.02987 | 0.01773 | 1.68475 | 0.09211 |
| sqrt($NO_2(t-1)$) | 0.74879 | 0.00951 | 78.75035 | 0.00000 |

The Tables A13–A15 show the calculated regression coefficients of the $\ln(PM_{10})$ linear regression models. The associated standard error (SE), t-statistic, and *p*-values are also shown in the tables for each model component. Tables A16–A18 show the same regression models with the autoregression (AR) $\ln(PM_{10}(t-1))$ component included as part of the model.

**Table A13.** Model 13: $\ln(PM_{10})$ linear regression model hourly data 2018–2019.

|  | Coefficients | SE | t-Stat | *p*-Value |
|---|---|---|---|---|
| Intercept | 3.94380 | 0.03402 | 115.93455 | 0.00000 |
| Traffic | 0.00015 | 0.00001 | 13.25308 | 0.00000 |
| Temperature | −0.02484 | 0.00135 | −18.45328 | 0.00000 |
| Precipitation | −0.51549 | 0.05868 | −8.78550 | 0.00000 |
| U Wind | 0.14868 | 0.01763 | 8.43297 | 0.00000 |
| V Wind | 0.17647 | 0.01236 | 14.27321 | 0.00000 |

**Table A14.** Model 14: $\ln(PM_{10})$ linear regression model hourly data 2020.

|  | Coefficients | SE | t-Stat | *p*-Value |
|---|---|---|---|---|
| Intercept | 3.84819 | 0.05599 | 68.73493 | 0.00000 |
| Traffic | 0.00018 | 0.00002 | 10.34412 | 0.00000 |
| Temperature | −0.03755 | 0.00252 | −14.88111 | 0.00000 |
| Precipitation | −0.44549 | 0.06350 | −7.01620 | 0.00000 |
| U Wind | 0.14197 | 0.02920 | 4.86140 | 0.00000 |
| V Wind | 0.15380 | 0.02142 | 7.18150 | 0.00000 |

**Table A15.** Model 15: $\ln(PM_{10})$ linear regression model hourly data 2018–2020.

|  | Coefficients | SE | t-Stat | *p*-Value |
|---|---|---|---|---|
| Intercept | 3.89823 | 0.02848 | 136.89227 | 0.00000 |
| Traffic | 0.00017 | 0.00001 | 18.22968 | 0.00000 |
| Temperature | −0.02939 | 0.00117 | −25.12901 | 0.00000 |
| Precipitation | −0.51918 | 0.04248 | −12.22137 | 0.00000 |
| U Wind | 0.15472 | 0.01500 | 10.31433 | 0.00000 |
| V Wind | 0.16793 | 0.01071 | 15.67994 | 0.00000 |

**Table A16.** Model 16: ln(PM$_{10}$) linear regression model hourly data 2018–2019 with auto-regression.

|  | Coefficients | SE | t-Stat | *p*-Value |
|---|---|---|---|---|
| Intercept | 0.80609 | 0.04559 | 17.68078 | 0.00000 |
| Traffic | 0.00007 | 0.00001 | 10.04389 | 0.00000 |
| Temperature | −0.00565 | 0.00088 | −6.40866 | 0.00000 |
| Precipitation | −0.17544 | 0.03719 | −4.71812 | 0.00000 |
| U Wind | 0.02990 | 0.01120 | 2.66922 | 0.00763 |
| V Wind | 0.03520 | 0.00799 | 4.40798 | 0.00001 |
| $ln(PM_{10}(t-1))$ | 0.77587 | 0.00995 | 77.93967 | 0.00000 |

**Table A17.** Model 17: ln(PM$_{10}$) linear regression model hourly data 2020 with auto-regression.

|  | Coefficients | SE | t-Stat | *p*-Value |
|---|---|---|---|---|
| Intercept | 1.12914 | 0.08701 | 12.97710 | 0.00000 |
| Traffic | 0.00009 | 0.00001 | 6.68386 | 0.00000 |
| Temperature | −0.01115 | 0.00197 | −5.66141 | 0.00000 |
| Precipitation | −0.12139 | 0.04673 | −2.59745 | 0.00949 |
| U Wind | 0.04073 | 0.02127 | 1.91486 | 0.05572 |
| V Wind | 0.04402 | 0.01576 | 2.79212 | 0.00531 |
| $ln(PM_{10}(t-1))$ | 0.69068 | 0.01957 | 35.28452 | 0.00000 |

**Table A18.** Model 18: ln(PM$_{10}$) linear regression model hourly data 2018–2020 with auto-regression.

|  | Coefficients | SE | t-Stat | *p*-Value |
|---|---|---|---|---|
| Intercept | 0.87330 | 0.04006 | 21.79908 | 0.00000 |
| Traffic | 0.00008 | 0.00001 | 12.43526 | 0.00000 |
| Temperature | −0.00683 | 0.00081 | −8.46584 | 0.00000 |
| Precipitation | −0.14594 | 0.02801 | −5.21053 | 0.00000 |
| U Wind | 0.03519 | 0.00987 | 3.56513 | 0.00037 |
| V Wind | 0.03760 | 0.00714 | 5.26758 | 0.00000 |
| $ln(PM_{10}(t-1))$ | 0.75641 | 0.00888 | 85.17302 | 0.00000 |

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
