# Peer review of "Air Pollution Modeling for Sustainable Urban Mobility with COVID-19 Impact Analysis: Case Study of Skopje"

_sustainability, doi:10.3390/su15021370_

Round 1
Reviewer 1 Report
Page 1, Title of the paper, “Traffic Pollution Modelling for Sustainable Urban Mobility with COVID-19 Impact Analysis: Case Study of Skopje”: My suggestion is to change the title to “Air Pollution Modelling for Sustainable Urban Mobility with COVID-19 Impact Analysis: Case Study of Skopje”. It is clear that you refer to air pollution due to traffic since the title includes “sustainable urban mobility”.
Page 1, Abstract, “…the full impact of transport on air quality is not…”: I do not fully agree with your statement. Please provide, within your manuscript, a detailed justification to support your statement.
Page 1, Abstract: Please include the main findings of your research at the end of the abstract.
Page 1, Keywords: Please change “Traffic Pollution” to “Air Pollution”.
Page 1, Keywords: Please include “COVID-19” in the keywords.
Section 1. Introduction: Please clearly state the research question (s) in the specific Section.
Section 2. Literature review: The literature review is rather weak (13 references only) considering the fact that the bibliography is very rich on the topic of your research. I kindly ask you to substantially increase the number of the references and the associated text within your manuscript.
Section 3. Materials and Methods: Please include a Data Flow Chart (DFC) describing all your methodological steps, for the benefit of the reader.
Page 3, subsection 3.1.1. Traffic flow dataset, line 103, “…traffic flows on four intersections…”: My suggestion is to include the selection criteria for the four intersections. Please also add some representative photographs, if possible.
Please insert “n.a.” in the empty cells of Tables 7, 8 and 9.
Section 5. Discussion: The specific Section includes the discussion of your findings only. Please investigate whether your findings comply or not with the respective findings at international level (e.g., case studies abroad) and if “yes”, to what extent? Therefore, I kindly ask you to improve Section 5.
Section 6. Conclusions: Please include the limitations and constraints of your research.
Section 6. Conclusions: Please include the policy recommendations arising from your findings. Please address each one of the recommendations to the respective stakeholders. To do so, please carry out a stakeholders’ analysis and include this analysis in your paper (who will benefit from your work and how).
Author Response
Dear reviewer, thank you for reviewing our paper. We managed to improve the manuscript according to your suggestions. In continuation we provide a detailed explanation and response for each of your suggestions:
Reviewer comment: Page 1, Title of the paper, “Traffic Pollution Modelling for Sustainable Urban Mobility with COVID-19 Impact Analysis: Case Study of Skopje”: My suggestion is to change the title to “Air Pollution Modelling for Sustainable Urban Mobility with COVID-19 Impact Analysis: Case Study of Skopje”. It is clear that you refer to air pollution due to traffic since the title includes “sustainable urban mobility”.
Author response: Thank you for your comment. We have changed the title of our paper to the one you suggested to highlight that the topic of our paper is indeed air pollution.
Reviewer comment: Page 1, Abstract, “…the full impact of transport on air quality is not…”: I do not fully agree with your statement. Please provide, within your manuscript, a detailed justification to support your statement.
Author response: Thank you for your comment. We agree that the statement is not entirely correct and we removed it from the abstract.
Reviewer comment: Page 1, Abstract: Please include the main findings of your research at the end of the abstract.
Author response: Thank you for your suggestions. We have added the our main findings to the end of the abstract section.
Reviewer comment: Page 1, Keywords: Please change “Traffic Pollution” to “Air Pollution”.
Author response: Thank you for your suggestion. We changed the keyword “Traffic Pollution” to “Air Pollution”.
Reviewer comment: Page 1, Keywords: Please include “COVID-19” in the keywords.
Author response: Thank you for your suggestion. We included the keyword “COVID-19” in the keywords list.
Reviewer comment: Section 1. Introduction: Please clearly state the research question (s) in the specific Section.
Author response: Thank you for your comment. The research question is now more clearly stated in the introduction section of the paper.
Reviewer comment: Section 2. Literature review: The literature review is rather weak (13 references only) considering the fact that the bibliography is very rich on the topic of your research. I kindly ask you to substantially increase the number of the references and the associated text within your manuscript.
Author response: Thank you for your comment. We increased the scope of our literature review extensively and added multiple references which include a review paper that summarizes the current state of research in this topic. In addition the literature review section now includes a detailed explanation of the identified research gap in the literature and the contribution of our paper.
Reviewer comment: Section 3. Materials and Methods: Please include a Data Flow Chart (DFC) describing all your methodological steps, for the benefit of the reader.
Author response: Thank you for your suggestion. We included a Data Flow Chart in section 3 of our paper to better describe all the methodological steps. In addition, the title of section 3 is now changed to Data and methodology as per suggestion of another reviewer.
Reviewer comment: Page 3, subsection 3.1.1. Traffic flow dataset, line 103, “…traffic flows on four intersections…”: My suggestion is to include the selection criteria for the four intersections. Please also add some representative photographs, if possible.
Author response: Thank you for your suggestion. The intersection selection criteria are now better explained in section 3. We could not obtain representative photographs of the selected intersections, but we updated the map showing their locations, and added more descriptions in subsection 3.1.1.
Reviewer comment: Please insert “n.a.” in the empty cells of Tables 7, 8 and 9.
Author response: Thank you for your suggestion. We inserted “n.a.” in all empty cells in Tables 7, 8 and 9.
Reviewer comment: Section 5. Discussion: The specific Section includes the discussion of your findings only. Please investigate whether your findings comply or not with the respective findings at international level (e.g., case studies abroad) and if “yes”, to what extent? Therefore, I kindly ask you to improve Section 5.
Author response: Thank you for your comment. We included a discussion in Section 5. on how our findings comply with other findings and case studies.
Reviewer comment: Section 6. Conclusions: Please include the limitations and constraints of your research.
Author response: Thank you for your suggestion. The limitations of our research are now clearly stated in the conclusion section.
Reviewer comment: Section 6. Conclusions: Please include the policy recommendations arising from your findings. Please address each one of the recommendations to the respective stakeholders. To do so, please carry out a stakeholders’ analysis and include this analysis in your paper (who will benefit from your work and how).
Author response: Thank you for your suggestion. Several policy recommendations are now included in the conclusion to the respective stakeholders.
Reviewer 2 Report
The authors showcased a series of temporal analysis on traffic’s, alongside with weather’s, impacts on air pollutant concentration using data from the capital city of North Macedonia, Skopje. Overall, the findings are not surprising given the number of published empirical studies with similar results. The technical aspect of this paper is more interesting, though, especially with a good display of modeling skills analyzing temporal dynamics of data.
There is such a contrast between the strong methodological sections and the weak empirical results that leads to my overall concern about this paper, which is, ‘what is your contribution to literature?’ In the introduction section, the authors merely point out the fact that the restrictive Covid-19 measures in North Macedonia ‘provided the opportunity for analysis’. To me, this is exactly what an ad hoc paper sounds like, i.e., we simply run some analysis because a natural experiment setting occurred due to Covid. There has to be a more convincing research core for this paper: What is the vital question to be answered? Is the question answered with the empirical analysis in this paper? This research core has to be put front and center in your introduction section.
I am okay with the literature review section except that the authors should highlight not only research findings but also research gaps. If there is not a gap to be addressed, what is the point of publishing this paper? Please refer to Silva, Branco, and Sousa (2022) for a more systematic review of literature on your topic and identify a research gap that may be addressed in your research.
Please change the Section 3 title to ‘Data and Methodology’. ‘Materials’ is not precise.’
Figure 1 is a poorly made map. Please add a sub-layer showing the relative location & the administrative boundary of Skopje in North Macedonia to give readers an idea about where the city is located. Also, add a legend showing the roadway classification, so the readers have an idea about what the traffic would look like. Add a north arrow to show the orientation of the map layer.
In Section 3.2.1 (p.5, line 149-151), please further elaborate on ‘changing driver behavior or traffic infrastructure changes’ that may explain the traffic trends for I2 and I4 that did not rebound to the normal levels like I1 and I3. For the subsection about the negative correlation between ozone and carbon monoxide concentration (p.6, line 178-182), some elaboration and references are needed. See Brancher (2021).
Section 3.2.2., while it is probably the most interesting part in this paper, is filled with jargon and heavy-lifting calculations that are obscure, even incomprehensible at times. For instance, what is stated in the Nyquist–Shannon sampling theorem and what it is used for (seems to justify the transformation of a discrete function into continuous one if I am not mistaken). There is also very little explanation of Figure 6 and Figure 7, which demonstrate the results of FFT. This part appear to be quite technical and marginally contribute to the overall discussion. I suggest that the authors move the two figures into the technical appendix and provide better explanations there. In addition, what it means to use ‘Gaussian smoothing with a window size of 200’ – what is the unit of analysis for this window size and what does 200 mean? I also feel confused about the terms ‘phase shifts’ and ‘fundamental frequencies’. They are probably some important properties of harmonic functions. Please define these terminologies to help readers understand what you try to convey.
In terms of the linear regression results, they are not surprising. The AR models performing better than the linear models is also expected. I wonder why you did not try to add an interaction term between traffic and the restrictive period indicator such that the coefficient directly reveal whether having a restriction affects the relationship between traffic and air pollutant concentration. It seems more direct to me. Alternatively, you should calculate the marginal effects of traffic on each air pollutant concentration between normal years and 2020 and compare them. Either way, I feel those are more direct comparisons than the regression coefficients themselves.
Lastly, the conclusion section is not particularly strong, but that has to do with an undefined research core in your introduction section. I believe once you make a stronger statement in the introduction section, you will be able to revise your conclusion section accordingly to reflect what the overall contribution of this paper is to literature (and the research topic).
Very solid writing!
References
Silva, A. C. T., Branco, P. T. B. S., & Sousa, S. I. V. (2022). Impact of COVID-19 Pandemic on Air Quality: A Systematic Review. International Journal of Environmental Research and Public Health, 19(4), 1950. https://doi.org/10.3390/ijerph19041950
Brancher, M. (2021). Increased ozone pollution alongside reduced nitrogen dioxide concentrations during Vienna’s first COVID-19 lockdown: Significance for air quality management. Environmental Pollution (1987), 284, 117153. https://doi.org/10.1016/j.envpol.2021.117153
Author Response
Dear reviewer, thank you for the insightful review of our paper. We managed to improve the manuscript according to your suggestions. In continuation we provide a detailed explanation and response for each of your suggestions:
Reviewer comment: There is such a contrast between the strong methodological sections and the weak empirical results that leads to my overall concern about this paper, which is, ‘what is your contribution to literature?’ In the introduction section, the authors merely point out the fact that the restrictive Covid-19 measures in North Macedonia ‘provided the opportunity for analysis’. To me, this is exactly what an ad hoc paper sounds like, i.e., we simply run some analysis because a natural experiment setting occurred due to Covid. There has to be a more convincing research core for this paper: What is the vital question to be answered? Is the question answered with the empirical analysis in this paper? This research core has to be put front and center in your introduction section.
Author response: Thank you for your comment and concerns. We began our traffic-air pollution modelling in the beginning of 2020 before COVID-19, and attempted to create models which would allow for some prediction of air quality in terms of traffic concentration. It was difficult to properly describe the influence on traffic as in Skopje specifically the air pollution is on a very high level due to its geographical position and presence of industry in the city. A natural experiment did occur then because of COVID-19 and allowed us to look into air pollution levels with drastically reduced traffic. A research question we asked ourselves was would our models before COVID-19 work the same way in 2020 during COVID (i.e. would their coefficients and R^2 value stay the same). Before COVID-19 such an event would be considered an outlier and probably removed from the analysis. We modified the paper in several sections to better emphasize the research question we are trying to answer.
Reviewer comment: I am okay with the literature review section except that the authors should highlight not only research findings but also research gaps. If there is not a gap to be addressed, what is the point of publishing this paper? Please refer to Silva, Branco, and Sousa (2022) for a more systematic review of literature on your topic and identify a research gap that may be addressed in your research.
Author response: Thank you for your comment. We expanded the literature review significantly, and then identified research gaps that are addressed in our paper. The Silva, Branco, and Sousa (2022) paper was most insightful in this regard.
Reviewer comment: Please change the Section 3 title to ‘Data and Methodology’. ‘Materials’ is not precise.
Author response: Thank you for your suggestion. We have changed the title of section 3 to ‘Data and Methodology’ as we agree that the term materials is not precise in the context of our paper.
Reviewer comment: Figure 1 is a poorly made map. Please add a sub-layer showing the relative location & the administrative boundary of Skopje in North Macedonia to give readers an idea about where the city is located. Also, add a legend showing the roadway classification, so the readers have an idea about what the traffic would look like. Add a north arrow to show the orientation of the map layer.
Author response: Thank you for your comment. We agree that the map was not clear enough. Given the time constraint we included a map of North Macedonia alongside the map of Skopje and added a north arrow to indicate the orientation of the map.
Reviewer comment: In Section 3.2.1 (p.5, line 149-151), please further elaborate on ‘changing driver behavior or traffic infrastructure changes’ that may explain the traffic trends for I2 and I4 that did not rebound to the normal levels like I1 and I3. For the subsection about the negative correlation between ozone and carbon monoxide concentration (p.6, line 178-182), some elaboration and references are needed. See Brancher (2021).
Author response: Thank you for your suggestions. We changed the sentence about changing driver behavior to emphasize that it is opinion of the authors that a possible redistribution of traffic flows occurred which resulted in reduced traffic flows in year 2020 for intersections I2 and I4. We consider further analysis of this phenomena beyond the scope of our paper, but agree that further investigation should be conducted. Regarding the negative correlation between ozone and carbon monoxide we explained their relation in more detail and included your suggested reference.
Reviewer comment: Section 3.2.2., while it is probably the most interesting part in this paper, is filled with jargon and heavy-lifting calculations that are obscure, even incomprehensible at times. For instance, what is stated in the Nyquist–Shannon sampling theorem and what it is used for (seems to justify the transformation of a discrete function into continuous one if I am not mistaken). There is also very little explanation of Figure 6 and Figure 7, which demonstrate the results of FFT. This part appear to be quite technical and marginally contribute to the overall discussion. I suggest that the authors move the two figures into the technical appendix and provide better explanations there. In addition, what it means to use ‘Gaussian smoothing with a window size of 200’ – what is the unit of analysis for this window size and what does 200 mean? I also feel confused about the terms ‘phase shifts’ and ‘fundamental frequencies’. They are probably some important properties of harmonic functions. Please define these terminologies to help readers understand what you try to convey.
Author response: Thank you for your comment. The goal of this part of the paper was to transfer our data from time-domain to a frequency domain. A look into the frequency domain allows for easier observation of temporal characteristics of data. When transferred to a frequency domain the data can be analyzed as a signal composed of multiple sine waves with different frequencies and shifts in phase. The Nyquist-Shannon sampling theorem states that the highest frequency we can observe is equal to twice the sample rate of our data in the time-domain (in our case sample rate was 1 sample per hour, which corresponds to a frequency of 0.5 [1/h]). After the signal is transformed to a frequency domain by FFT and presented in the form of a periodogram (figure 6 and 7) analysis of frequencies with high amplitudes can be conducted. A periodogram shows power spectral densities which shows how the signal strength depends on frequency. In our data we encountered high amplitudes on frequencies which correspond to 24, 12, 8, 6 hours. We consider this important as it shows the cyclical nature of both traffic and air pollution. Gaussian smoothing was used on the periodograms (figure 6 and 7) to allow for easier visual inspection of the results for the benefit of the reader. The window of 200 means that the periodograms were smoothed by taking the median of 200 data points, there is no unit for this value as it is just a number of data points. We would prefer it if the periodograms stay in this section and not be moved to the appendix. The terms fundamental frequency and phase shift are now better explained in the paper.
Reviewer comment In terms of the linear regression results, they are not surprising. The AR models performing better than the linear models is also expected. I wonder why you did not try to add an interaction term between traffic and the restrictive period indicator such that the coefficient directly reveal whether having a restriction affects the relationship between traffic and air pollutant concentration. It seems more direct to me. Alternatively, you should calculate the marginal effects of traffic on each air pollutant concentration between normal years and 2020 and compare them. Either way, I feel those are more direct comparisons than the regression coefficients themselves.
Author response: Thank you for your comment. We did not add an interaction term between traffic and restrictive period because our goal was to see if there was any change in linear regression model coefficients between years 2018, 2019 and 2020. The restrictive period was rather short in the total amount of data to draw conclusions by just looking at that segment in the data. Marginal effects are by definition the partial derivative of the regression function. Since our model is simple and uses only linear terms the partial derivatives would be equal to regression coefficients themselves.
Reviewer comment: Lastly, the conclusion section is not particularly strong, but that has to do with an undefined research core in your introduction section. I believe once you make a stronger statement in the introduction section, you will be able to revise your conclusion section accordingly to reflect what the overall contribution of this paper is to literature (and the research topic).
Author response: Thank you for your comment. We agree that it is imperative to focus on the core of our research in the introduction and write a conclusion that reflects the main contributions of this paper. We have modified the conclusion section along with the previously mentioned introduction to better reflect upon this.
Reviewer comment: Very solid writing!
References
Silva, A. C. T., Branco, P. T. B. S., & Sousa, S. I. V. (2022). Impact of COVID-19 Pandemic on Air Quality: A Systematic Review. International Journal of Environmental Research and Public Health, 19(4), 1950. https://doi.org/10.3390/ijerph19041950
Brancher, M. (2021). Increased ozone pollution alongside reduced nitrogen dioxide concentrations during Vienna’s first COVID-19 lockdown: Significance for air quality management. Environmental Pollution (1987), 284, 117153. https://doi.org/10.1016/j.envpol.2021.117153
Author response: Thank you for your comment and suggested literature. We found both references enlightening and they are now included in the manuscript reference list.
Round 2
Reviewer 1 Report
I would like to express my deepest thanks to the authors because they have carefully addressed my comments. However, there is one change that needs to be made in reference 14 (Dimovska, M.; Gjorgjev, D. The Effects of COVID-19 Lockdown on Air Quality in Macedonia. Open Access Macedonian Journal of 486, Medical Sciences 2020, 8, 353–362. https://doi.org/10.3889/oamjms.2020.5455.) Please change “Macedonia” to “North Macedonia” since this is the official name of the country.
Author Response
Dear reviewer,
thank you for your comment and kind words. We agree that the name of the country is "North Macedonia" and this is why in our manuscript we used this name in our text. However title of the cited paper (https://doi.org/10.3889/oamjms.2020.5455.) used only the name Macedonia. Since this is the title of the published paper by other authors and not us we are not sure if we are at liberty to change it in our reference list. Hence, we will leave this decision to the editor.
The authors